# Influence of Inorganic Additives on the Surface Characteristics, Hardness, Friction and Wear Behavior of Polyethylene Matrix Composites

**DOI:** 10.3390/ma16144960

**Published:** 2023-07-12

**Authors:** Natalia Wierzbicka, Rafał Talar, Karol Grochalski, Adam Piasecki, Wiesław Graboń, Miłosz Węgorzewski, Adam Reiter

**Affiliations:** 1Faculty of Mechanical Engineering, Poznan University of Technology, 60-965 Poznan, Poland; rafal.talar@put.poznan.pl (R.T.); karol.grochalski@put.poznan.pl (K.G.);; 2Faculty of Materials Engineering and Technical Physics, Poznan University of Technology, 60-965 Poznan, Poland; adam.piasecki@put.poznan.pl; 3Faculty of Computer Science, Rzeszow University of Technology, 35-959 Rzeszow, Poland; wgrabon@prz.edu.pl

**Keywords:** polyethylene, hBN, titanium, friction, surface analysis

## Abstract

The aim of this research was to analyze the effect of inorganic additives on the tribological properties of the high-density polyethylene (HDPE) matrix composite surface. Titanium (Ti) and hexagonal boron nitride (hBN) were added in different mass fractions. The samples were produced by pressing a pre-prepared mixture of granules. The composite samples with the following mass fractions of additives were fabricated: 5% hBN, 10% hBN, 28% Ti–2% hBN, 23% Ti–7% hBN, and 20% Ti–10% hBN. An even distribution of individual additives’ concentrations was confirmed. Observations of morphology, surface topography, hardness, and tribological measurements were conducted using reciprocating motion tests with the “pin-on-flat” and rotational tests with the “pin-on-disc” configuration. Subsequently, microscopic observations and measurements of the wear track profile were carried out. Additionally, geometry parameters of the contacting elastic body were calculated for various counter-samples. It was found that the Shore D hardness of samples containing Ti and hBN increased with the Ti content, while the coefficient of friction (COF) value decreased. The addition of hBN alone did not significantly affect the hardness, regardless of the ratio, while the COF increased with the increasing hBN content. The COF value doubled with the addition of 10% hBN (COF = 0.22), whereas the addition of 90% Ti–10% hBN resulted in a decrease in the COF value, to COF = 0.83. The highest hardness value was obtained for the sample containing 28% Ti–2% hBN (66.5), while the lowest was for the sample containing 10% hBN (63.2). The wear track analysis, including its height and width caused by deformation, was detected using a focal differentiation microscope and scanning electron microscopy. Additionally, EDS maps were generated to determine the wear characteristics of the composite.

## 1. Introduction

The selection of an appropriate polymer material, as well as the shape and dimensions of the sliding elements, is a crucial aspect in the design of polymers for tribological applications. Before choosing a polymeric material for friction-related applications, specific application conditions need to be taken into consideration. Currently, a wide range of polymers and their composites are utilized in the production of sliding elements for various industrial applications. Recent studies have focused on various aspects of the sliding surface, including tribological investigations of different polymers and the role of mating surfaces [1]. Additionally, the literature provides results of coefficient of friction (COF) measurements for various polymers employed as plain bearings [2,3]. Enhancing tribological parameters and mechanical properties remains a popular research topic. For instance, Di Maro et al. [4] and Baduruthamal et al. [5] have published works on modifying high-density polyethylene (HDPE) with alumina-toughened zirconia (ATZ) and carbon nanotubes (CNTs). Aliyu et al. proposed the utilization of graphene to improve the tribological properties of ultra-high-molecular-weight polyethylene (UHMWPE) in their study [6].

The use of composites comprising high-resistance polymers filled with heat-conducting powders, such as titanium or copper metal powder, bronze, aluminum, etc., helps overcome the problem of high-temperature bonding. The conducted tests and analyses of tribological properties facilitate the evaluation of the material’s properties and its potential range of applications, following the development of the initial samples. According to Hutchings and Shipway [7], this allows for obtaining initial data on the material’s suitability for its intended purpose at an early stage of the project, enabling the selection of optimal lubricants or the determination of susceptibility to surface wear.

The study of composite materials is an important field due to the wide range of possibilities it offers in many aspects. It has been found that a properly designed and manufactured composite can serve as an excellent alternative to existing materials by reducing the consumption of scarce raw materials. By combining a smaller amount of these materials with a less expensive one, it becomes possible to change the manufacturing process to a less energy-intensive or less environmentally polluting one [8]. These factors directly contribute to the reduction of technological costs, supporting the development of enterprises. Moreover, as emphasized by Knight and Curliss [9], composite materials facilitate technological advancements by providing the means necessary to implement purely theoretical ideas. 

In this specific case, the samples analyzed in a tribological context were prepared to assess the influence of hexagonal boron nitride (hBN) and titanium (Ti) on the properties of pure high-density polyethylene (HDPE). 

HDPE is a suitable base material for applications such as water-lubricated crankshaft bearings due to its environmentally friendly properties, low cost, stable chemical properties, and good lubricity. Under low-speed and high-load conditions, boundary lubrication and direct-contact friction occur within the wear area between the bearing and the shaft contact [4]. The characteristics of hBN were investigated by Lipp et al. [10], while the self-lubricating properties of hBN were confirmed by Kuang et al. [11]. Nguyen et al. explored the effect of TiO_2_ nanocomposites on the properties of polyethylene [12].

The particle size is of great importance. In their study, Wang et al. [13] investigated the influence of hBN particles on the properties of PEO coatings. In terms of wear resistance, coatings containing smaller hBN particles exhibited higher strength compared to those containing larger particles. The fine particles acted as solid lubricants, reducing friction and wear during sliding contact. This effect can be attributed to the formation of a self-lubricating tribofilm consisting of hBN particles and oxide phases on the coating surface. Additionally, the presence of smaller hBN particles improved the corrosion resistance of the PEO coatings.

Therefore, samples containing additives were prepared to effectively reduce surface wear while maintaining low friction coefficient values. The conducted tests and analyses of tribological properties facilitate the evaluation of the material’s properties and its potential range of applications after the initial sample development. Improving the tribological properties of high-density polyethylene (HDPE) through the use of inorganic additives is a highly relevant topic. The following are several articles that discuss the influence of additives on the mechanical properties of composites. In their study, Mahmoud et al. [14] utilized inorganic MgO and ZnO nanoparticles in a high-density polyethylene (HDPE) matrix and investigated their impact on enhancing the chemical, physical, morphological, structural, and mechanical properties compared to the neat polymer. Zhang et al. [15] examined the influence of graphene nanoplatelets (GNP) on the tribological properties of high-density polyethylene (HDPE) based on friction experiments. The results showed that GNP exhibited a limited friction reduction effect on HDPE, which could be controlled by the mass content of GNP, load, and speed. Wu and his team [16] prepared a new composite material consisting of high-density polyethylene (HDPE) and multi-walled carbon nanotubes (MWCNT) for water-lubricated stern tube bearings, which play a significant role in the marine propulsion system. The results indicated that MWCNT significantly improved the mechanical properties of HDPE and weakened deformation behaviors at the friction interface, resulting in a reduction in COF and wear volume under both dry and water-lubricated conditions.

The aim of this study was to examine the mechanical properties of composites based on high-density polyethylene (HDPE) with different mass fractions of inorganic additives, namely Ti and hBN, and to assess their impact on material wear. The additives used in this study had relatively large grain sizes. However, in the analyzed studies, additives with much smaller grain sizes were employed. After reviewing the literature, no information was found regarding the influence of these additives used individually or in combination on HDPE-based composites. The chosen method for composite fabrication in this study was the cost-effective pressing technique, without the use of any activators.

## 2. Materials and Methods

### 2.1. Plan of Experiments

This study aimed to assess the essential mechanical properties of polyethylene composites that have been modified with additives. Furthermore, the impact of these modifications on the tribological behavior and surface topography was investigated as a supplementary part of the experimental analysis. The conducted tests encompassed tribological examinations, including the determination of the coefficient of friction and the analysis of pathway deformation.

### 2.2. Materials

The samples were prepared using high-density polyethylene (referred to as “PE” for convenience) as the base material, and they were enriched with different mass fractions of hexagonal boron nitride (hBN) and/or titanium (Ti) (as shown in Table 1). The choice of titanium and boron nitride (hBN) as fillers for the polyethylene matrix can be dictated by several factors. Both titanium and boron nitride have a beneficial impact on the hardness, strength, and tribological properties of the composite, including the sliding properties of hBN, which reduce friction and improve lubrication between surfaces. The addition of these fillers increases wear resistance, reduces friction, and minimizes wear, as well as enhances high-temperature resistance. Furthermore, the chemical compatibility of titanium and boron nitride with the polyethylene matrix facilitates their incorporation into the composite and maintains structural stability. Additionally, both titanium and boron nitride are non-toxic and can be safely used in contact with humans.

The powders were obtained from ABCR GmbH. According to the manufacturer, the titanium (Ti) powder had a purity of 99.7% by weight, while the hexagonal boron nitride (hBN) powder had a purity of 99% by weight, with impurities of 0.3% and 1% by weight, respectively. To verify the composition, the powder morphology was examined, and photographs were captured using the Tescan MIRA3 Scanning Electron Microscope (Figure 1).

During the production of the samples, relatively large particles of inorganic materials, such as titanium (Ti) and hexagonal boron nitride (hBN), were utilized. This choice was influenced by their availability and cost-effectiveness. Larger-sized Ti and hBN particles are more readily accessible and economical compared to their nanoscale counterparts. This is advantageous for large-scale production or industrial applications where cost considerations are significant. Furthermore, the use of larger particles simplifies the processing techniques associated with fabricating polyethylene (PE) nanocomposites. Achieving dispersion and incorporation of larger particles into the PE matrix is relatively easier using conventional processing methods. Although smaller nanoparticles typically exhibit enhanced reinforcement due to their larger surface area, larger particles can still provide some level of reinforcement. The larger particle size contributes to improved mechanical properties, such as increased stiffness and strength, albeit to a lesser extent compared to nanoscale fillers. The choice of using larger particles depends on the application requirements. In certain cases, larger particles may be preferred to optimize specific characteristics, including thermal conductivity, electrical resistance, or wear resistance, based on the intended application of the PE nanocomposites.

### 2.3. Sample Preparation

The samples were fabricated using a pressing technique with a pre-prepared material. The process involved several steps. Firstly, a mixture of powders and granules was extruded, followed by grinding and repeating this process three times to ensure thorough mixing of the material. Next, the sample was compressed using a mold designed for round samples measuring 55 mm in diameter and 8 mm in height. The compression was carried out at a temperature of approximately 180 °C. Subsequently, finishing operations were performed, including turning, drilling mounting holes, and polishing the samples. The manufacturing process is illustrated in Figure 2.

### 2.4. Element Concentration Distribution

The surfaces of the samples and wear tracks were analyzed using the EDS method. The distribution of element concentrations, including PE, Ti, and hBN, was studied using the Ultim Max 65 microanalyzer (Oxford Instruments, High Wycombe, UK).

### 2.5. Surface Topography Test

The surface topography of the samples was analyzed using a contact profilometer, specifically the Hommel T8000 (Hommelwerke GmbH, Villingen-Schwenningen, Germany). The tests were performed in accordance with ISO 25178 standards [17], and measurements were taken for various surface characteristics, including roughness, waviness, surface-bearing capacity, structure directivity, and an approximate image of the surface depicting unevenness dimensions. Each test was repeated five times on the front surface of the samples. The roughness parameter was quantified as the root mean square height (Sq). This parameter provides a three-dimensional representation of the profile parameter, Rq, indicating the root mean square for Z(x, y) within the evaluation area [18]:(1)$$Sq=\sqrt{\frac{1}{A}\overset{n}{\underset{A}{\iint }}Z^{2}\left(x,y\right)\;dxdy}$$

### 2.6. Hardness Test

The hardness of the samples was determined using a Shore Durometer (Sauter GmbH, Wutöschingen, Germany). Measurements were conducted at approximately 3 mm intervals on the surfaces of the samples. The reported measurement result represents the average value of 10 measurements.

### 2.7. Tribological Tests

Tribological tests were conducted using the BRUKER UMT2 Tribolab tester (Bruker Corporation, Billerica, MA, USA) in both pin-on-flat and pin-on-disc configurations. The tests were performed according to the relevant standards (G133-05 and ASTM G99) [19,20].

In the pin-on-flat configuration, a ball made of 1.2378 steel with a diameter of 10 mm and a roughness of Ra = 0.6 μm was in continuous sliding contact with the composite specimen. The testing methodology was described by Czapczyk et al. [21]. In the pin-on-disc configuration, a pin with a diameter of 25 mm was used.

The tribological test in the pin-on-flat configuration was conducted for a duration of 20 min with a reciprocating displacement amplitude of 30 mm and a maximum velocity of 25 mm/s. The tests were performed with three different force values: 5 N, 10 N, and 15 N. The force value of 10 N was selected for further experiments due to the similar values of the coefficient of friction obtained in all three measurements. Additionally, measurements with a force of 10 N showed the most stable results. In the pin-on-disc configuration, comparative tests were carried out with a force of 10 N, a duration of 30 min, and a maximum velocity of 15 mm/s. The coefficient of friction was evaluated using Equation (2) [22,23]:μ = (A_r τ)/W(2)
where μ represents the friction coefficient, A_r denotes the actual contact area, τ signifies the effective shear strength of the contacts, and W represents the applied normal load, in Newtons (N).

For these testing methods, the tribotester was equipped with a fixed and rotating table. The samples were clamped to immobilize them and prevent any movement during the measurements. 

Each composite was subjected to at least three parallel experiments to ensure adequate statistical evaluation. The average value of the coefficient of friction for each sample is presented along with the corresponding standard deviations. The tests were conducted at room temperature without the use of lubricants.

### 2.8. Geometry of Contacting Elastic Body

In addition to utilizing advanced technology, valuable insights can also be gained through calculations. By considering parameters such as the load (expressed as W (in N)) and the radius of the measuring ball (R (in m)), it is possible to determine the stress values that occur during the test. These calculations can be based on the formulas provided by Stachowiak [24]:(3)$$\frac{1}{R_{x}}=\frac{1}{R_{ax}}+\frac{1}{R_{bx}}$$
(4)$$\frac{1}{R_{y}}=\frac{1}{R_{ay}}+\frac{1}{R_{by}}$$
where R_ax_ and R_ay_ represent the radius of curvature of the measuring ball, which are equal, and R_bx_ and R_by_ refer to the curvature of the surface of the tested object, which can be considered approximately infinite in this case.

This relationship satisfies the condition $\frac{1}{R_{x}}=\frac{1}{R_{y}}$, allowing the use of the formula for the reduced radius of curvature:(5)$$\frac{1}{R^{\prime }}=\frac{1}{R_{x}}+\frac{1}{R_{y}}$$

The reduced Young’s modulus, as described in the work of N’Jock [25], is necessary for further calculations:(6)$$\frac{1}{E^{\prime }}=\frac{1-v_{1}^{2}}{E_{1}}+\frac{1-v_{2}^{2}}{E_{2}}\;\left[\mathrm{MPa}\right]$$
where:$v_{1},v_{2}$—Poisson’s ratio for materials 1 and 2,$E_{1},E_{2}$—Young’s modulus of materials 1 and 2 (MPa).

These data allow for the determination of the contact area between the measuring ball and the test sample:(7)$$a={\left(\frac{3WR^{\prime }}{E^{\prime }}\right)}^{\frac{1}{3}}\;[m]$$
where:W—normal load (N).

Subsequently, the maximum and average pressure on the object can be calculated:(8)$$P_{max}=\frac{3W}{2\pi a^{2}}\;\left[\mathrm{MPa}\right]$$
(9)$$P_{average}=\frac{W}{\pi a^{2}}\;\left[\mathrm{MPa}\right]$$

The maximum deflection can be determined as:(10)$$\delta =1.0397{\left(\frac{W^{2}}{E^{\prime 2}R^{\prime }}\right)}^{\frac{1}{3}}\;\left[m\right]$$

The maximum shear stress is given by:(11)$$\tau_{max}=\frac{1}{3}P_{max}\;\left[\mathrm{MPa}\right]$$

The depth at which the maximum shear stress occurs is approximately:(12)z=0.638a [m]

By obtaining these calculated values, it becomes possible to simulate the loads applied to the samples during tribological tests and adjust them based on the strength characteristics of the tested materials.

### 2.9. Wear Track Observation

Detailed observations were carried out using an Alicona Infinite Focus UR-10 optical profilometer (Bruker Alicona, Graz, Austria) and scanning electron microscopy (SEM) with a Tescan MIRA3 microscope (Tescan, Czech Republic, Brno, Kohoutovice) at magnifications of 200×. These examinations enabled the analysis of the precise deformation of the dynamically loaded surface of the samples. The study aided in elucidating the wear mechanism, obtaining an average track profile, and measuring its depth and width.

## 3. Results and Discussion

### 3.1. Element Concentration Distribution

After preparing the samples, the distribution of element concentrations was measured to verify whether the individual components were evenly distributed and properly mixed. The tests were conducted on a microsection of a sample containing PE, Ti, and hBN. The distribution map of the element concentrations (at approximately 100 µm) confirmed that the assumption was met and the samples were correctly produced (Figure 3). Additionally, the picture confirmed the presence of particles: N, B, Ti, and the material (C).

To further confirm the presence of N and B, a test with a closer approximation to 5 µm was performed (Figure 4). These results were qualitative but confirmed their presence. 

### 3.2. Surface Topography

The tribological properties were influenced by surface topography, as confirmed in previous studies [26]. Therefore, the surface was examined to determine whether it was properly prepared for further testing. The obtained results, presented in Table 2, varied depending on the proportion of additives. Polyethylene samples containing hBN were easily machinable, as indicated by the low values of the Sq parameter. On the other hand, the addition of Ti significantly impeded the finishing work. The addition of hBN to PE slightly decreased the parameter value, while the presence of Ti had no significant effect on the obtained values and the surface quality.

### 3.3. Hardness Measurements

The hardness values of the investigated samples are presented in Figure 5. A clear increase in Shore D values was observed depending on the Ti content. The addition of hBN to pure PE led to a slight decrease in the hardness value.

The impact of the addition of Ti and hBN on the change in hardness for samples containing both additives is analyzed in Figure 6. In the case of samples with hBN (Figure 6A), the hardness initially slightly decreased with increasing content, and then increased again. The addition of Ti and hBN resulted in a significant increase in hardness. Hardness increased by about 3% with the decreasing hBN content and the increasing Ti (Figure 6B).

### 3.4. Evaluation of Tribological Properties

The friction coefficient charts were automatically generated from tribological tests using the computer software integrated with the BRUKER UMT2 TriboLab device. It was observed that, for all samples, the friction coefficient value reached a stable state after a certain distance (Figure 7).

Initially, the coefficient of friction was relatively low and gradually increased until it reached a steady state. This behavior is commonly observed in polymers [27,28] and is attributed to the adhesion of the polymer to the steel counter-sample. During the test, a thin layer of polymer, known as a transfer film, can be deposited on the opposing sample [29]. Samples with the addition of hBN exhibited a longer stabilization time compared to other samples. Various factors, including the properties of the outer layer and the extent of surface deformation caused by friction, contributed to the overall stabilization time of the coefficient of friction.

In Figure 8, the influence of hBN on PE and PE-Ti is presented. The addition of only 10% hBN resulted in a 72% increase in the coefficient of friction (COF) for 90PE–10hBN, while a decrease of 10% was observed for 70PE–20Ti–10hBN compared to pure polymer. Excessive hBN content can lead to the formation of clusters, which hinders the self-lubricating action. Additionally, a high concentration of the powder may result in weak bonding with the polymer matrix, causing an increase in the coefficient of friction. The incorporation of titanium contributed to a decrease in the COF, which could be attributed to reduced adhesive forces and increased hardness.

Figure 9 presents the steady-state values of the coefficient of friction for PE samples with various additive compositions. The addition of hBN to PE resulted in an increase in the COF value, while samples containing Ti and hBN reached lower values than the baseline. The COF value increased with the increase of hBN content and the decreasing Ti content. The lowest value was recorded for the sample containing 28% wt. Ti and 2% wt. hBN, while the highest value was observed for the sample with 25% wt. Ti and 5% wt. hBN. 

The increase in the coefficient of friction (COF) with the increase in hBN content can be attributed to several factors. hBN, being a material with lubricating properties, can reduce the friction between surfaces, resulting in a lower COF. However, as the hBN content increases, particle agglomeration may occur, leading to an irregular distribution of the filler and potentially increasing friction and wear. Additionally, a higher filler content can result in increased stiffness and less elastic behavior of the material, which can increase the point contact and contribute to higher friction. Such a phenomenon may explain the increase in the COF value.

The addition of Ti to the PE-hBN composite can have different effects on the COF depending on the proportions of Ti and hBN. In the case of PE-Ti, Ti can act as a filler, influencing the structure and mechanical properties of the composite. In smaller amounts, Ti can contribute to reducing the friction and wear by forming a protective layer on the surface, reducing adhesion, and decreasing surface contact. However, as the Ti content becomes higher, the opposite effect may occur. A higher Ti content can lead to more agglomerates and a greater impact on the flexibility and stiffness of the composite. This phenomenon can lead to an increase in the coefficient of friction.

However, the introduction of Ti as an additive can affect the resulting COF in unexpected ways. Several factors may contribute to this effect. Firstly, Ti can alter the interactions between hBN particles and the polymer matrix. Ti can form chemical bonds or surface interactions with the polymer, leading to increased adhesion between the filler phase and the matrix. This increased adhesion can elevate the friction between surfaces and result in a higher COF. Secondly, differences in the composition ratios of Ti and hBN can lead to different interactions between these additives. In the case of the COF of 70PE–23Ti–7hBN and 70PE–20Ti–10hBN, variations in the proportions of Ti and hBN can result in different interactions between these components and the polymer matrix. These differences can influence the surface structure, elasticity, and adhesion, which can translate into different COF outcomes.

Analyzing the graph in Figure 10, which shows the value of the friction coefficient as a function of hBN content, for samples with the addition of hBN (Figure 10A), the COF value increased to 0.21 (a 43% increase) with the increase in hBN content. In Figure 10B, samples containing both additives are analyzed. A constant COF value (10% lower than PE) was observed for an hBN content of 10% wt. (20% wt. Ti) and 7% wt. (23% wt. Ti). A significant decrease of about 20% in COF occurred in the case of 3% wt. (70PE–27Ti–3hBN).

In addition, tests were carried out with a counter-sample with a diameter of 25 mm in rotation. The steady-state values of the coefficient of friction for selected samples are presented in Figure 11. The tests were conducted on four selected samples. The highest value was recorded for the addition of 5% wt. hBN, while the lowest was observed for samples with the addition of 10% wt. hBN. The differences in COF values were small (up to 0.01), which was attributed to low stresses and a larger contact surface compared to the case of a 10 mm-diameter ball.

### 3.5. Geometry Parameter Calculations of Contacting Elastic Body

With polymeric materials, pressure and stress play a crucial role, making calculations related to contact geometry essential. These calculations provide proper characterization of the tested materials and insight into the wear mechanism. Table 3 presents the calculations performed for the polyethylene–steel pair. The values were computed for tests conducted with a 10 N pressure using a ball with radii of 10 mm and 25 mm. It can be observed that a larger contact area resulted in a reduced impact on the tested surface, leading to significantly lower coefficient of friction (COF) values and narrower wear paths. Increasing the diameter of the counter-sample decreased the occurring stresses by 8% and the maximum strains by 17%. Moreover, lower stresses and a larger contact area resulted in relatively unstable tribological tests, with less discrepant values. The lack of measurement stability was caused by insufficient point pressure, ball bouncing, and sliding of the counter-sample.

### 3.6. Microscope Observation

Following the completion of the tribological tests, microscopic observations were conducted using a differential focus microscope and scanning electron microscopy (SEM). The results are presented in Table 4. In the case of the base sample (PE), high adhesive forces between the ball and the sample were observed, resulting in material exfoliation and significant elastic deformation. After adding hBN (95PE–5hBN and 90PE–10hBN), increased deformations were observed, which may be attributed to the increased hardness and deterioration of the elastic properties of the composite. Analyzing the SEM images, a change in surface wear was evident. The addition of hBN and an increase in its content resulted in reduced surface wear. However, traces of finishing machining are still visible, which may be attributed to the self-lubricating properties of the additive. In the remaining samples (with the addition of Ti and hBN), the titanium particles led to reduced adhesive forces, and despite the increased hardness, significant deformations were not observed. The wear track after the tribological examination of samples with Ti and hBN additives exhibited a more defined shape, and the surface layer was noticeably worn, indicating even poorer elastic properties compared to samples with hBN alone.

Figure 12 illustrates the wear track profile for all tested samples. It can be observed that samples with hBN alone and Ti with hBN additives exhibited a more well-defined profile of the wear track compared to the base sample. The elastic properties of pure PE minimized the sample’s deformation. Based on this, it can be concluded that the additives led to a deterioration of the polymer’s elastic properties.

Table 5 presents the values for the geometric parameters of the wear track, namely width and depth. The values were averaged from 1000 profiles. The base material (PE) exhibited the smallest depth but the largest width. The addition of Ti resulted in a decrease in the width value to 718.65 µm for 70PE–23Ti–2hBN and 6.17 µm for 70PE–28Ti–2hBN.

Figure 13 displays the changes in the width and depth values for the tested materials. The additives caused a decrease in the width of the wear path, possibly due to reduced adhesion compared to the base sample. The depth values for the remaining samples significantly increased compared to PE, which can be attributed to the increased hardness and reduced flexibility of the composites. With the increasing hBN content, the depth decreased, indicating the lubricating effect of the hBN particles. However, an exception was observed for 70PE–28Ti–2hBN, where the depth value was the largest, possibly due to the insufficient hBN addition compared to Ti.

### 3.7. EDS Results

The results of the EDS analysis conducted on the wear track after the tribological tests are presented in Table 6. In the case of samples containing only the hBN additive, a uniform distribution of the additive was observed. However, for a higher content (10% by weight), the presence of powder clusters was noticed, which may affect the increase in the COF value. Analyzing the images of samples containing both Ti and hBN additives, titanium clusters were highly visible, which affected the COF value. In the case of the 70PE–28Ti–2hBN sample, a small addition of hBN (at 28% by weight of Ti) positively affected the wear, as it was the lowest. This could be attributed to the increased hardness while utilizing the self-lubricating properties of hBN. The presence of titanium particles throughout the surface and their dispersion was also observed. The remaining two samples (70PE–23Ti–7hBN and 70PE–20Ti–10hBN) were characterized by the presence of hBN clusters, and the presence of Ti on the entire surface was less visible compared to the previously discussed 70PE–28Ti–2hBN sample. 

The presented images revealed the presence of individual Ti and hBN particles embedded within the HDPE matrix. Ti particles appeared as small, dispersed clusters on the surface, while hBN particles exhibited a more uniform distribution. This distribution pattern suggests that Ti particles have a greater tendency to aggregate compared to hBN particles.

## 4. Conclusions

This study addressed the impact of inorganic additives, namely hBN and Ti, of a relatively large size, on the mechanical and tribological properties of HDPE. The Ti additive was characterized by high hardness and the formation of a protective layer on the surface, reducing friction and wear. hBN acted as a lubricating additive, reducing friction and improving wear resistance. The combination of Ti and hBN in polyethylene resulted in a synergistic effect, enhancing the tribological properties. Ti influenced hardness, while hBN influenced lubrication.

The presence of both additives with evenly distributed elemental concentrations was confirmed. The hBN additive improved the surface properties by reducing the Sq parameter, while the Ti and hBN additives increased it. The addition of Ti increased the hardness, while hBN had no significant effect on this value. Samples containing Ti and hBN showed decreased COF values with the increasing hardness. Conversely, an increase in hBN content led to an increase in the COF values. The behavior of the COF for samples containing Ti and hBN exhibited variability over time.

Inorganic additives reduce the surface wear and impact the crack depth. The hBN additive demonstrated wear–reducing properties, while the Ti additive reduced the adhesive forces. For samples containing Ti and hBN, the material became less elastic, more brittle, and the contact area decreased. These samples showed the lowest COF values. Specifically, the addition of 10% hBN resulted in a 72% increase in COF for 90PE–10hBN, while a decrease of 10% was observed for 70PE–20Ti–10hBN compared to the pure polymer.

The use of hBN enhanced the resistance to deformation of the wear track. When comparing this phenomenon among three types of samples (PE, PE–hBN, and PE–Ti–hBN), more pronounced surface deformations were observed for the samples containing PE–Ti–hBN. The width of the analyzed wear track decreased with the increase in the hBN content, which may be attributed to the formation of clusters. Additives increased the depth of the wear track, deteriorating the material’s elastic properties. EDS maps in the wear track revealed the presence of individual Ti and hBN particles embedded within the HDPE matrix. Ti particles appeared as small, dispersed clusters on the surface, while hBN particles exhibited a more uniform distribution. This distribution pattern suggests that Ti particles have a greater tendency to aggregate compared to hBN particles, which is also influenced by the additive content.

Calculating the geometric parameters for the contacting elastic body enabled the determination of the influence of the counter-sample size on the contact surface, as well as the associated stresses and strains. 

In the future, research is planned to be conducted for additives with smaller grain sizes.

## Figures and Tables

**Figure 1 materials-16-04960-f001:**
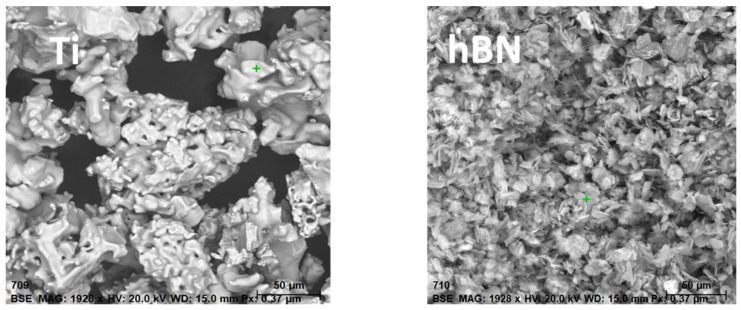
Powder morphology obtained using the Tescan MIRA3 Scanning Electron Microscope.

**Figure 2 materials-16-04960-f002:**
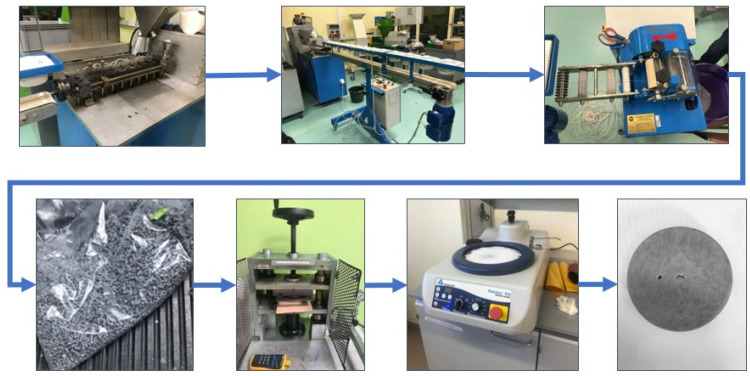
Sample production process.

**Figure 3 materials-16-04960-f003:**
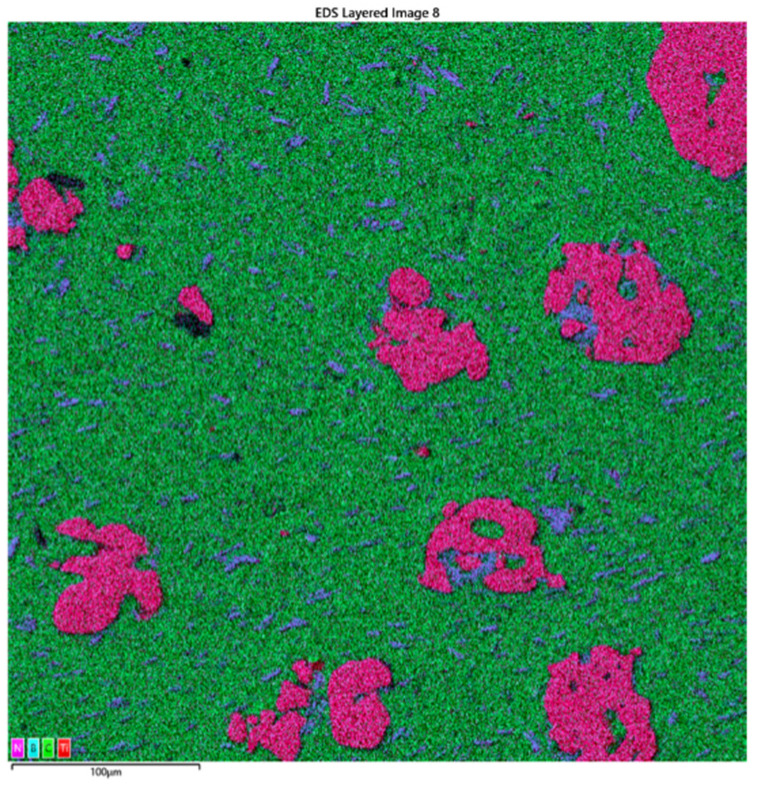
Map of the distribution of elemental concentrations for the micro-section of the sample obtained using the EDS Ultim Max 65 X-ray microanalyzer (approximately 100 µm).

**Figure 4 materials-16-04960-f004:**
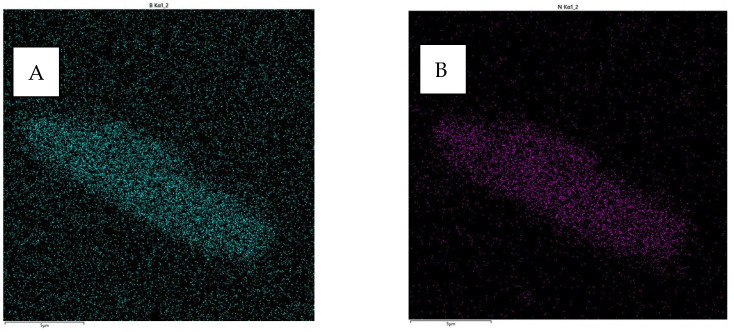
Map of the distribution of elemental concentrations for the cross-section of the sample obtained using the EDS Ultim Max 65 X-ray microanalyzer (approximately 5 µm). (**A**) B particles and (**B**) N particles.

**Figure 5 materials-16-04960-f005:**
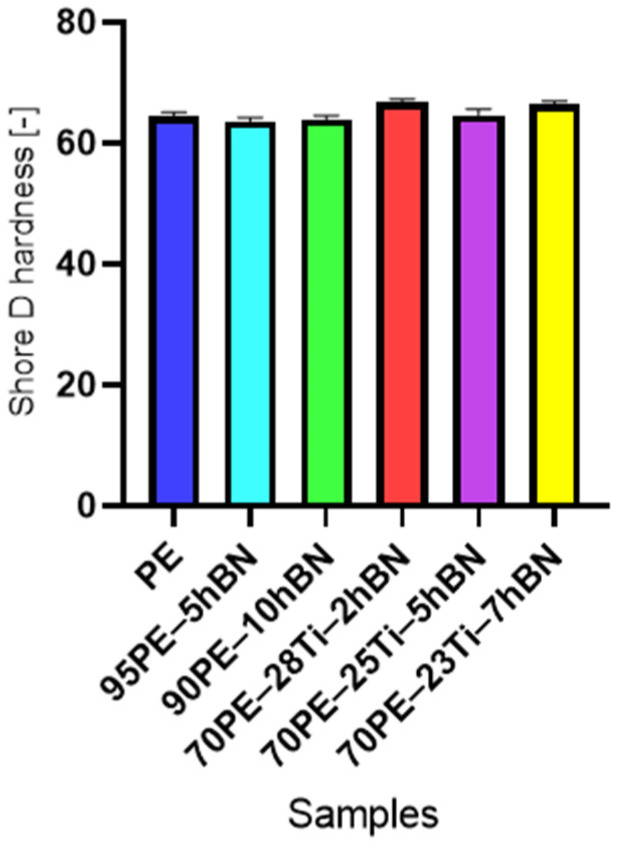
The Shore D hardness of the samples with various content compositions, produced by pressing.

**Figure 6 materials-16-04960-f006:**
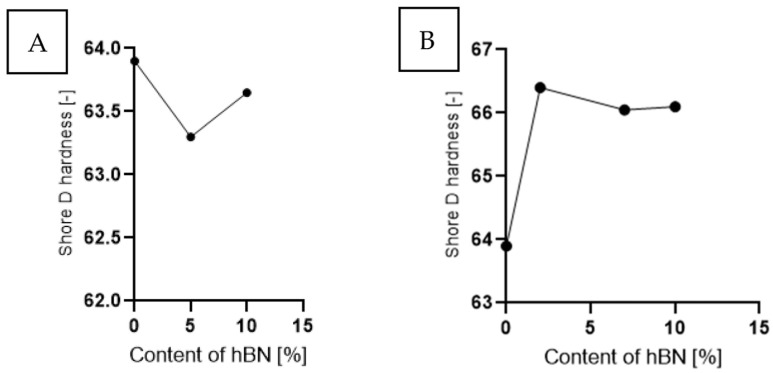
The Shore D hardness as a function of the hBN for samples containing: (**A**) hBN and (**B**) Ti and hBN.

**Figure 7 materials-16-04960-f007:**
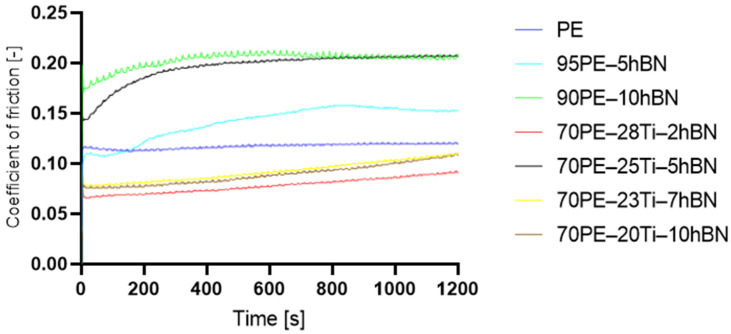
The coefficient of friction as a function of time for the samples with the normal load of 10 N.

**Figure 8 materials-16-04960-f008:**
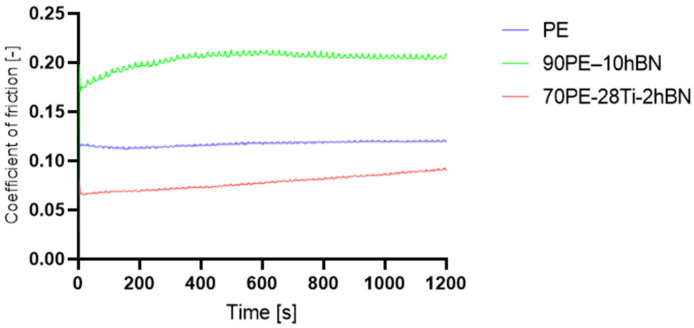
The coefficient of friction as a function of the sliding distance for the samples with the addition of 10% hBN with the normal load of 10 N.

**Figure 9 materials-16-04960-f009:**
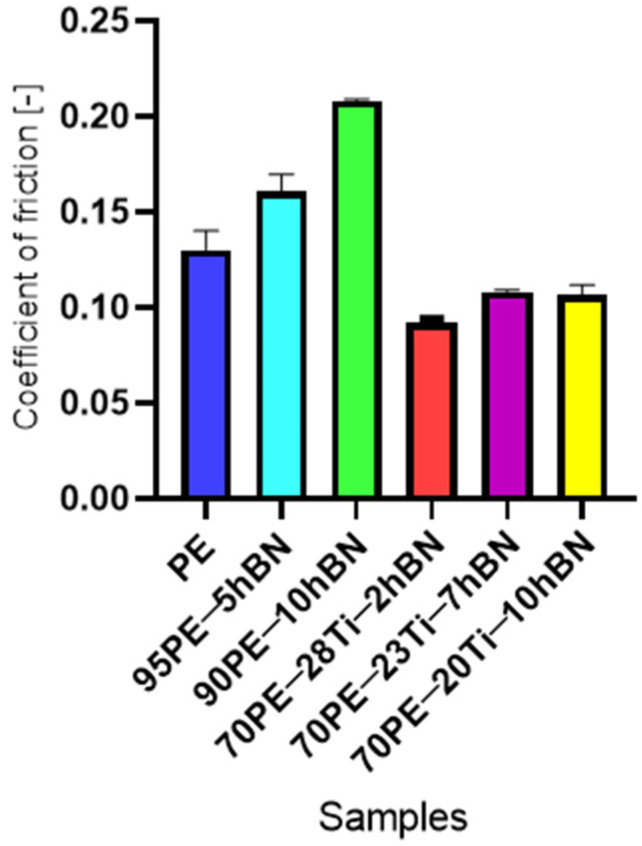
The steady-state values of the coefficient of friction of the PE samples with various additive compositions.

**Figure 10 materials-16-04960-f010:**
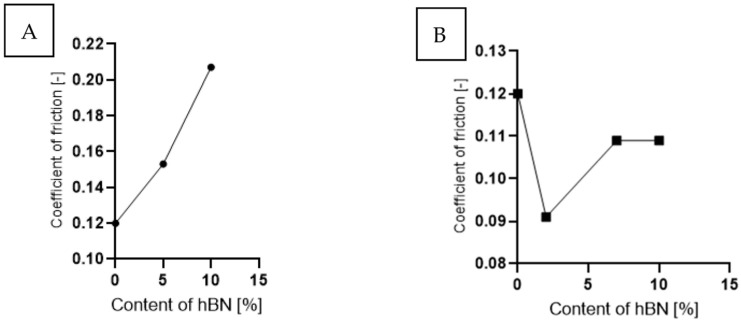
The coefficient of friction as a function of the hBN for samples containing: (**A**) hBN and (**B**) Ti and hBN.

**Figure 11 materials-16-04960-f011:**
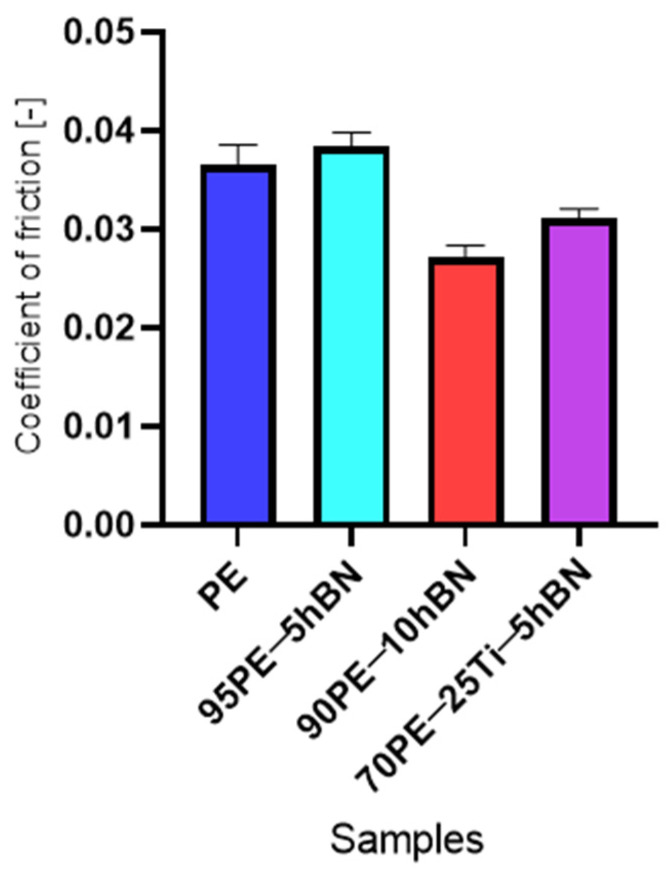
The steady-state values of the coefficient of friction of the PE samples with various additive compositions for a counter-sample with a diameter of 25 mm.

**Figure 12 materials-16-04960-f012:**
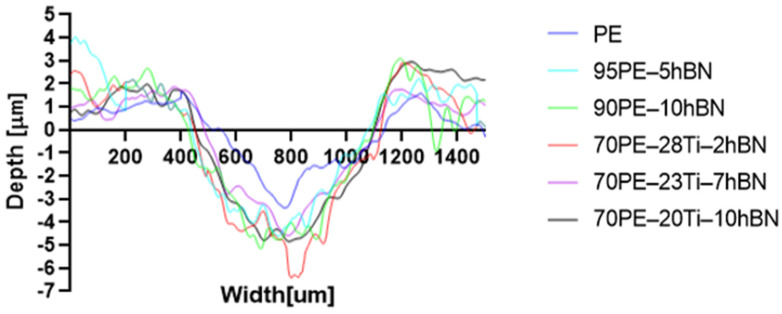
Wear track profile for all tested samples.

**Figure 13 materials-16-04960-f013:**
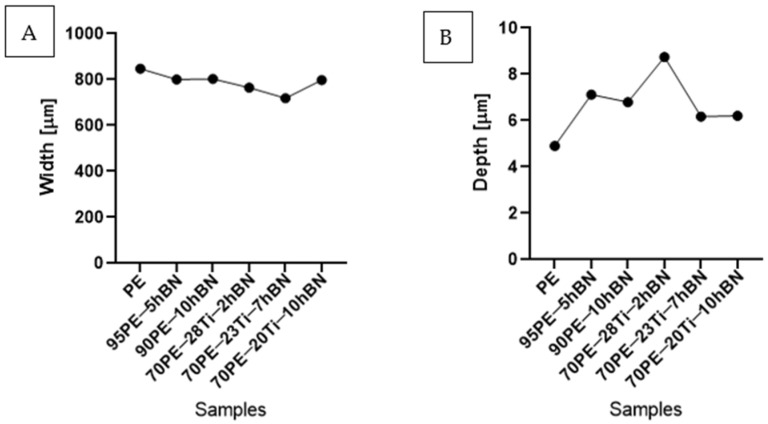
The changes in the width (**A**) and depth (**B**) values for the tested materials.

**Table 1 materials-16-04960-t001:** Content of additives in the tested samples.

Notation	Quantity of Additive (%)		
	PE	Ti	hBN
95PE–5hBN	95	-	5
90PE–10hBN	90	-	10
70PE–28Ti–2hBN	70	28	2
70PE–23Ti–7hBN	70	23	7
70PE–20Ti–10hBN	70	20	10

**Table 2 materials-16-04960-t002:** Surface topography map obtained using a Hommel T8000 contact profilometer.

Notation	Maps	Sq Parameter (μm)
PE	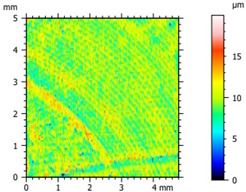	1.67
95PE–5hBN	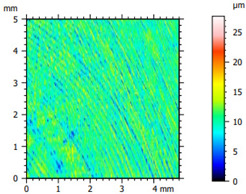	1.63
90PE–10hBN	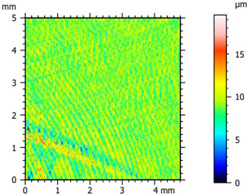	1.4
70PE–28Ti–2hBN	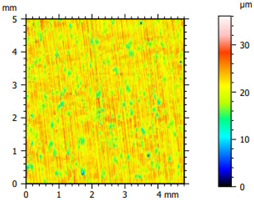	2.49
70PE–23Ti–7hBN	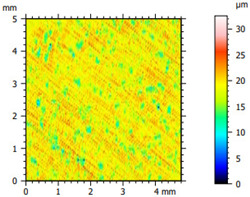	2.49
70PE–20Ti–10hBN	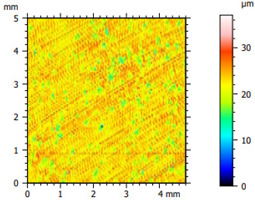	2.63

**Table 3 materials-16-04960-t003:** Geometry parameter calculations of the contacting elastic body.

Force	Diameter	Contact Area Dimensions	Maximum Contact Pressures	Average Contact Pressures	Maximum Deflection	Maximum Shear Stress	Depth at Which Maximum Shear Stress Occurs
F (N)	d (mm)	a (μm)	Pmax (MPa)	Pavg (MPa)	δ (μm)	τmax (MPa)	z (μm)
10	10	624.602	12.239	8.159	420.114	4.079	398.496
10	25	672.832	10.547	7.031	389.996	3.516	429.267

**Table 4 materials-16-04960-t004:** Comparison of the friction area as observed by the focal differentiation microscope and scanning electron microscopy.

Notation	Differential Focus Microscope Observation	Scanning Electron Microscopy Observation
PE	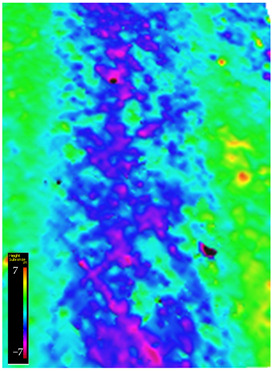	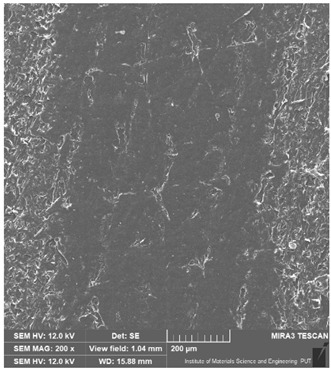
95PE–5hBN	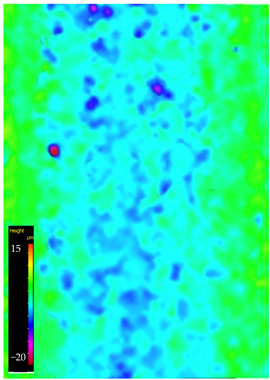	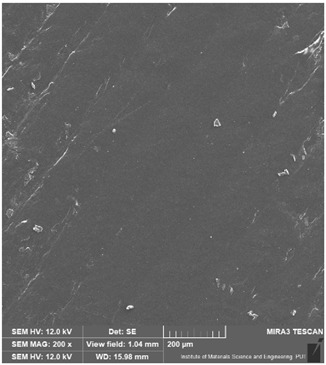
90PE–10hBN	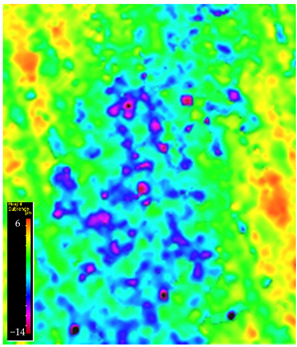	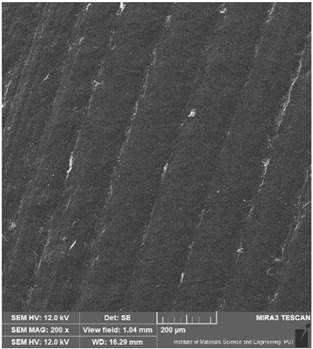
70PE–28Ti–2hBN	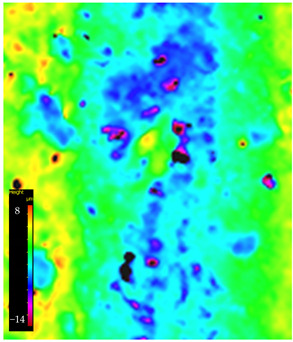	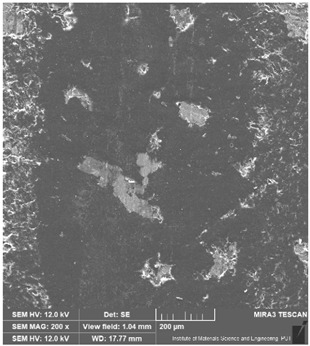
70PE–23Ti–7hBN	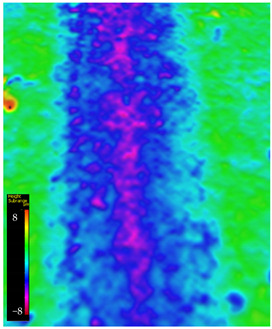	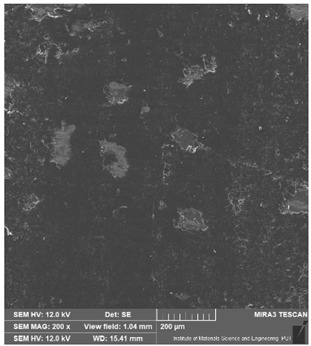
70PE–20Ti–10hBN	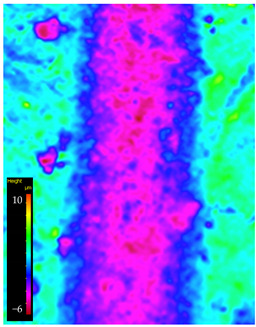	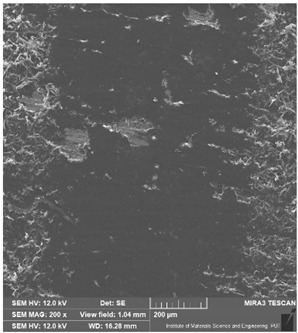

**Table 5 materials-16-04960-t005:** The geometrical parameters of the wear track profile.

Sample	Width (µm)	Depth (µm)
PE	847.05	4.9
95PE–5hBN	800.1	7.12
90PE–10hBN	802.52	6.79
70PE–28Ti–2hBN	764.71	8.74
70PE–23Ti–7hBN	718.65	6.17
70PE–20Ti–10hBN	798.04	6.2

**Table 6 materials-16-04960-t006:** Maps of the elemental concentration distribution in the wear path of the samples were obtained using the EDS Ultim Max 65 X-ray microanalyzer, with a spatial resolution of approximately 250 µm.

Notation	EDS Results
95PE–5hBN	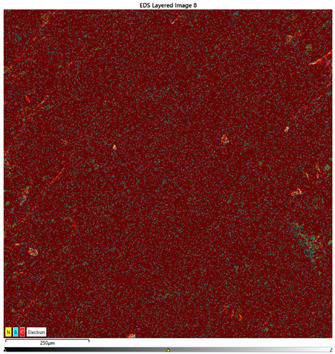
90PE–10hBN	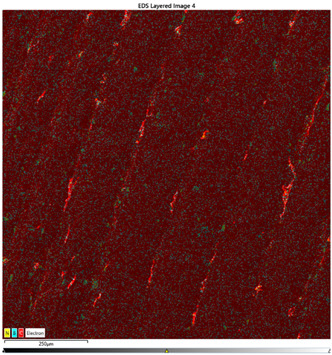
70PE–28Ti–2hBN	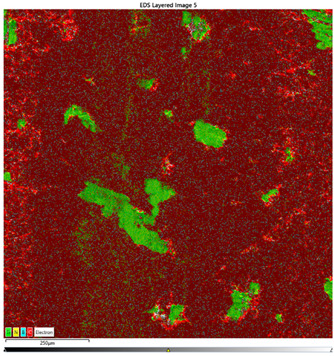
70PE–23Ti–7hBN	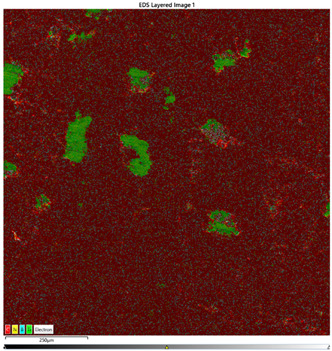
70PE–20Ti–10hBN	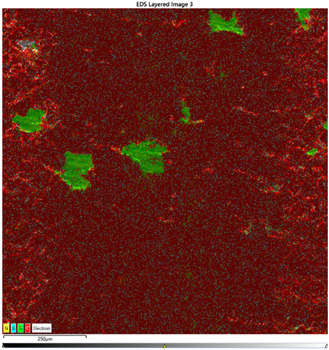

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
