# Peer review of "Influence of Inorganic Additives on the Surface Characteristics, Hardness, Friction and Wear Behavior of Polyethylene Matrix Composites"

_materials, 2023, doi:10.3390/ma16144960_

Round 1
Reviewer 1 Report
The effects of inorganic additives on the surface tribological properties of polyethylene matrix composites were analyzed. Different addition amounts of Ti and hBN were studied. The authors' research makes sense, but some improvements are needed before it can be accepted.
1. The authors in the introduction part cite too few literatures, and need to add literatures related to the author's research.
2. The author's interpretation of Figure 3 is not clear enough. What is the difference between the two graphs in Figure 3?
3. Figure 4 is not obvious enough to make clear what the author analyzes.
Section 4.3.3 There are some differences in surface topography, but they are relatively small, and the author needs to mark them in the figure.
5. The friction coefficient analysis in Figure 8 is not clear enough, so the author needs to focus on the analysis.
Overall, the author's study has some significance, but it needs further modification.
English needs moderate modification.
Author Response
Good Morning, I attach the answers and the manuscript.
- The authors in the introduction part cite too few literatures, and need to add literatures related to the author's research.
The literature has been added.
[13].Wang, Q.; Wang, S.; Du, N.; Cheng, F.;Zhao, Q.; Li, X. Effects of HBN particle sizes on microstructure and perfor-mances of PEO coatings on 7075-T6 aluminum alloy. Zhongguo Youse Jinshu Xuebao/Chinese Journal of Nonferrous Metals. 2019, 29, 11, 2459 – 24701.
[14].Mahmoud, M., E.; Khalifa, M., A.; Youssef, M., R.; El-Sharkawy, R., M. Influence of MgO and ZnO as nano-additives on the mechanical, microstructural and thermal performance of high-density polyethylene. Journal of Applied Polymer Science. 2022, 139, 3115.
[15].Liyuan, Z.; Conglin, D.; Chengqing, Y.; Xiuqin, B.; Ye T. The role of graphene nanoplatelets in the friction reducing process of polymer. Polymer Composites. 2022, 43, 11, 8213 – 8227.
[16].Wu, Y.; Dong, C.; Yuan, C.; Bai, X.; Zhang, L.; Tian, Y. MWCNTs filled high-density polyethylene composites to im-prove tribological performance. Wear. 2021, 47718, 203776.
- The author's interpretation of Figure 3 is not clear enough. What is the difference between the two graphs in Figure 3?
There is no difference, in the article there are two photos that show the same thing - one graph has been left, which shows the even distribution of elemental particles.
- Figure 4 is not obvious enough to make clear what the author analyzes.
Figure 4 - the photos confirm the presence of elements B and N in the tested samples, which are an addition to the composite. In figure 3, N is marked in purple, while at higher magnification in Figure 4, the presence of element B can also be seen in this place - for this purpose, this photo was included in the article
- Section 4.3.3 There are some differences in surface topography, but they are relatively small, and the author needs to mark them in the figure.
I am sorry but I do not understand what you mean - were you referring to the identification of clusters of additive grains in the images?
- The friction coefficient analysis in Figure 8 is not clear enough, so the author needs to focus on the analysis.
The analysis has been completed.
In Figure 8, the influence of hBN on PE and PE-Ti is presented. The addition of only 10% hBN resulted in a 72% increase in the coefficient of friction (COF) for 90PE-10hBN, while a decrease of 10% was observed for 70PE-20Ti-10hBN compared to pure polymer. Excessive hBN content can lead to the formation of clusters, which hinders the self-lubricating action. Additionally, a high concentration of the powder may result in weak bonding with the polymer matrix, causing an increase in the coefficient of friction. The incorporation of titanium contributed to a decrease in the COF, which could be attributed to reduced adhesive forces and increased hardness.
Overall, the author's study has some significance, but it needs further modification.
The indicated errors have been corrected, the analysis extended by description and additional research.
Comments on the Quality of English Language: English needs moderate modification.
The English language has been improved

Reviewer 2 Report
In this manuscript, the authors reported an investigation of the effect of inorganic additives on the tribological properties of the polyethylene matrix composite surface. Both titanium and hexagonal boron nitride were added into PE with different contents to obtain the composites. Thereafter, the authors examined the surface morphologies and mechanical properties of the composites. The results reported in this manuscript could interest the readers in the community of composites. This manuscript can be considered by this journal. However, there are several major problems which must be fixed before this manuscript can be re-considered:
1. The sizes of titanium (Ti) and hexagonal boron nitride (hBN) particles were too large, which resulted in a sharp decrease in the mechanical properties of the materials. The results of morphologies indicate that that inorganics materials did not have fine good dispersion in PE. There have been a great number of reports on the preparation of the nanocomposites of PE with the inorganic nanoparticles. The authors should justify the use of the inorganic particles with so big sizes!
2. The coefficient of friction for PE-hBN increased with increasing the contents of hBN. Upon introducing Ti, the COF decreased significantly. However, the COF of 70PE-23Ti-7hBN was higher than 70PE-20Ti-10hBN, which is opposite to the result of increasing Ti. Please explain this!
3. It is hard to see the tendency of the change in wear for the composites of PE with Ti-hBN as functions of the contents of the inorganics according to the geometrical parameters such as width and depth. The authors over explained the mechanism of wear properties only with the very limited data.
4. The authors failed to explain and discuss the experimental data. This is a descriptive paper and the discussion must be strengthened. It lacks the explanations of mechanism behind the results.
Moderate editing of English language required
Author Response
Good Morning, I attach the answers and the manuscript.
- The sizes of titanium (Ti) and hexagonal boron nitride (hBN) particles were too large, which resulted in a sharp decrease in the mechanical properties of the materials. The results of morphologies indicate that that inorganics materials did not have fine good dispersion in PE. There have been a great number of reports on the preparation of the nanocomposites of PE with the inorganic nanoparticles. The authors should justify the use of the inorganic particles with so big sizes!
The justification for the use of inorganic particles of such large sizes has been substantiated.
During the production of the samples, relatively large particles of inorganic materials, such as titanium (Ti) and hexagonal boron nitride (hBN), were utilized. This choice was influenced by their availability and cost-effectiveness. Larger-sized Ti and hBN particles are more readily accessible and economical compared to their nanoscale counterparts. This is advantageous for large-scale production or industrial applications where cost considerations are significant. Furthermore, the use of larger particles simplifies the processing techniques associated with fabricating polyethylene (PE) nanocomposites. Achieving dispersion and incorporation of larger particles into the PE matrix is relatively easier using conventional processing methods. Although smaller nanoparticles typically exhibit enhanced reinforcement due to their larger surface area, larger particles can still provide some level of reinforcement. The larger particle size contributes to improved mechanical properties, such as increased stiffness and strength, albeit to a lesser extent compared to nanoscale fillers. The choice of using larger particles depends on the application requirements. In certain cases, larger particles may be preferred to optimize specific characteristics, including thermal conductivity, electrical resistance, or wear resistance, based on the intended application of the PE nanocomposites.
- The coefficient of friction for PE-hBN increased with increasing the contents of hBN. Upon introducing Ti, the COF decreased significantly. However, the COF of 70PE-23Ti-7hBN was higher than 70PE-20Ti-10hBN, which is opposite to the result of increasing Ti. Please explain this!
The COF for 70PE-23Ti-7hBN and 70PE-20Ti-10hBN was similar, with a lower value observed for 70PE-28Ti-2hBN.
Figure 9 presents the steady-state values of the coefficient of friction for PE sam-ples with various additive compositions. The addition of hBN to PE results in an in-crease in the COF value, while samples containing Ti and hBN reach lower values than the baseline. The COF value increases with the increase of hBN content and decreasing Ti content. The lowest value is recorded for a sample containing 28% wt. Ti and 2% wt. hBN, while the highest value is observed for samples with 25% wt. Ti and 5% wt. hBN.
The increase in the coefficient of friction (COF) with the increase in hBN content can be attributed to several factors. HBN, being a material with lubricating properties, can reduce the friction between surfaces, resulting in a lower COF. However, as the hBN content increases, particle agglomeration may occur, leading to an irregular dis-tribution of the filler and potentially increasing friction and wear. Additionally, a higher filler content can result in increased stiffness and less elastic behavior of the material, which can increase point contact and contribute to higher friction. Such a phenomenon may explain the increase in the COF value.
The addition of Ti to the PE-hBN composite can have different effects on the COF depending on the proportions of Ti and hBN. In the case of PE-Ti, Ti can act as a filler, influencing the structure and mechanical properties of the composite. In smaller amounts, Ti can contribute to reducing friction and wear by forming a protective layer on the surface, reducing adhesion, and decreasing surface contact. However, as the Ti content becomes higher, the opposite effect may occur. A higher Ti content can lead to more agglomerates and a greater impact on the flexibility and stiffness of the compo-site. This phenomenon can lead to an increase in the coefficient of friction.
However, the introduction of Ti as an additive can affect the resulting COF in un-expected ways. Several factors may contribute to this effect. Firstly, Ti can alter the in-teractions between hBN particles and the polymer matrix. Ti can form chemical bonds or surface interactions with the polymer, leading to increased adhesion between the filler phase and the matrix. This increased adhesion can elevate the friction between surfaces and result in a higher COF. Secondly, differences in the composition ratios of Ti and hBN can lead to different interactions between these additives. In the case of COF 70PE-23Ti-7hBN and 70PE-20Ti-10hBN, variations in the proportions of Ti and hBN can result in different interactions between these components and the polymer matrix. These differences can influence surface structure, elasticity, and adhesion, which can translate into different COF outcomes.
- It is hard to see the tendency of the change in wear for the composites of PE with Ti-hBN as functions of the contents of the inorganics according to the geometrical parameters such as width and depth. The authors over explained the mechanism of wear properties only with the very limited data.
The study was complemented with SEM photos of the wear track and EDS maps. The analysis was extended by description
- The authors failed to explain and discuss the experimental data. This is a descriptive paper and the discussion must be strengthened. It lacks the explanations of mechanism behind the results.
The study was complemented with SEM photos of the wear track and EDS maps. The analysis was extended by description
Moderate editing of English language required
The English language has been improved

Reviewer 3 Report
REVIEW
Review of the Manuscript MATERIALS-2458720 entitled “Influence of inorganic additives on the surface characteristics, hardness, friction and wear behavior of polyethylene matrix composites” by Natalia Wierzbicka, RafaÅ‚ Talar, Karol Grochalski, Adam Piasecki, WiesÅ‚aw GraboÅ„, MiÅ‚osz WÄ™gorzewski and Adam Reiter to be considered for publication in the JOURNAL OF MATERIALS MDPI.
In this paper, the authors studied the tribological characteristics of polyethylene composites with Ti and hBN. The samples were produced by pressing a granule mix. Morphology, topography, hardness, and tribological measurements in a pin-on-flat test and microscopy observations and wear track profile were carried out.
From their results, the authors report that the addition of hBN alone does not significantly affect the hardness but the friction coefficient increases. They indicate that inorganic additives tested reduce the wear and affect the depth of the cracks. The addition of hBN improved the surface properties by lowering the Sq parameter, while the addition of Ti and hBN increased it.
After reading this paper there are some questions for the authors:
· The abstract looks good. Please include all significant numerical results.
· For the introduction section, please add more references (related to the topic of the article (2021-2023)) and briefly explain them.
· What is the problem? Why was the manuscript written? Please explain the reason in the introduction part. In the last paragraph of the introduction, it should be expressed the novelty of the study and the differences from the past in detail.
· Improve the conclusion parts.
1) What exactly is the scientific novelty of your work?
2) What is the role of the Ti in the hardness and the influence of both hBN and Ti in the tribological performance of the polymer matrix? Can you explain in deeper detail the influence of the inorganic additives and not just describing the results? For example Why Ti influence friction response and wear?
In the materials and methods section, please provide the number of wear tests that you performed per condition.
Please provide EDS maps of the wear tracks.
Have you looked at the counter body worn surface? Have you observed material transfer? Please provide some images of the counter body.
Minor editing of English language required
Author Response
Good Morning, I attach the answers and the manuscript.
- The abstract looks good. Please include all significant numerical results.
The significant numerical results has been added
The aim of the research was to analyze the effect of inorganic additives on the tribological properties of the High-Density Polyethylene (HDPE) matrix composite surface. Titanium (Ti) and hexagonal boron nitride (hBN) were added in different mass fractions. The samples were produced by pressing a pre-prepared mixture of granules. The composite samples with the following mass fractions of additives were fabricated: 5% hBN, 10% hBN, 28% Ti-2% hBN, 23% Ti-7% hBN, and 20% Ti-10% hBN. An even distribution of individual additives' concentrations was confirmed. Observations of morphology, surface topography, hardness, and tribological measurements were conducted using reciprocating motion tests with the "pin on flat" and rotational tests with the "pin on disc" configuration. Subsequently, microscopic observations and measurements of the wear track profile were carried out. Additionally, geometry parameters of the contacting elastic body were calculated for various counter-samples. It was found that the Shore D hardness of samples containing Ti and hBN increases with the Ti content, while the coefficient of friction (COF) value decreases. The addition of hBN alone does not significantly affect the hardness, regardless of the ratio, while the COF increases with increasing hBN content. The COF value doubled with the addition of 10% hBN (COF = 0.22), whereas the addition of 90% Ti-10% hBN resulted in a decrease in the COF value to COF = 0.83. The highest hardness value was obtained for the sample containing 28% Ti-2% hBN (66.5), while the lowest was for the sample containing 10% hBN (63.2). The wear track analysis, including its height and width caused by deformation, was detected using a focal differentiation microscope and scanning electron microscopy. Additionally, EDS maps were generated to determine the wear characteristics of the composite.
- For the introduction section, please add more references (related to the topic of the article (2021-2023)) and briefly explain them
The literature has been added.
[13].Wang, Q.; Wang, S.; Du, N.; Cheng, F.;Zhao, Q.; Li, X. Effects of HBN particle sizes on microstructure and perfor-mances of PEO coatings on 7075-T6 aluminum alloy. Zhongguo Youse Jinshu Xuebao/Chinese Journal of Nonferrous Metals. 2019, 29, 11, 2459 – 24701.
[14].Mahmoud, M., E.; Khalifa, M., A.; Youssef, M., R.; El-Sharkawy, R., M. Influence of MgO and ZnO as nano-additives on the mechanical, microstructural and thermal performance of high-density polyethylene. Journal of Applied Polymer Science. 2022, 139, 3115.
[15].Liyuan, Z.; Conglin, D.; Chengqing, Y.; Xiuqin, B.; Ye T. The role of graphene nanoplatelets in the friction reducing process of polymer. Polymer Composites. 2022, 43, 11, 8213 – 8227.
[16].Wu, Y.; Dong, C.; Yuan, C.; Bai, X.; Zhang, L.; Tian, Y. MWCNTs filled high-density polyethylene composites to im-prove tribological performance. Wear. 2021, 47718, 203776.
- What is the problem? Why was the manuscript written? Please explain the reason in the introduction part. In the last paragraph of the introduction, it should be expressed the novelty of the study and the differences from the past in detail.
The aim of the study was to examine the mechanical properties of composites based on high-density polyethylene (HDPE) with different mass fractions of inorganic additives, namely Ti and hBN, and to assess their impact on material wear. The additives used in this study had relatively large grain sizes. However, in the analyzed studies, additives with much smaller grain sizes were employed. After reviewing the literature, no information was found regarding the influence of these additives used individually or in combination on HDPE-based composites. The chosen method for composite fabrication in this study was the cost-effective pressing technique, without the use of any activators.
- Improve the conclusion parts.
1) What exactly is the scientific novelty of your work?
The novelty of this study lies in the use of large grain sizes and the absence of any activator. Additionally, I could not find any information in the literature regarding the combination of the materials discussed.
The study addresses the impact of inorganic additives, namely hBN and Ti, of relatively large size, on the mechanical and tribological properties of HDPE. The Ti additive is characterized by high hardness and the formation of a protective layer on the surface, reducing friction and wear. HBN acts as a lubricating additive, reducing fric-tion and improving wear resistance. The combination of Ti and hBN in polyethylene results in a synergistic effect, enhancing the tribological properties. Ti influences hardness, while hBN influences lubrication.
2) What is the role of the Ti in the hardness and the influence of both hBN and Ti in the tribological performance of the polymer matrix? Can you explain in deeper detail the influence of the inorganic additives and not just describing the results? For example Why Ti influence friction response and wear?
Information on the impact of additives has been supplemented in the conclusions
The Ti particles have a significant influence on the hardness of the polyethylene matrix and its tribological properties. Ti is a material with relatively high hardness, which contributes to the increased hardness of the polymer matrix. The increased hardness improves the composite's resistance to wear and deformation under tribological loads. Ti can form a protective layer on the contact surface, reducing friction and wear. This occurs by the formation of oxidized layers on the surface, which can limit adhesion and provide lubrication between surfaces. Additionally, Ti can influence chemical reactions occurring on the surface, such as reactions with oxygen, further reducing friction and wear during the tribological process.
HBN acts as a lubricating additive, with the ability to reduce friction between contacting surfaces in the composite material. Due to its layered structure, HBN can also decrease wear by forming a lubricating layer on the contact surface. Furthermore, HBN can improve resistance to wear and tensile strength, resulting in enhanced tribological properties of the composite.
The combination of both HBN and Ti in the polyethylene matrix leads to a synergistic effect, where their combined presence contributes to even better tribological properties of the composite. The combination of the lubricating properties of HBN with the protective action of Ti can further reduce friction and wear, improving the durability and tribological performance of the material.
- In the materials and methods section, please provide the number of wear tests that you performed per condition.
The information is provided in the article. For the surface topography measurement, the study was repeated five times, for the hardness measurement it was repeated ten times, and for the tribological tests it was conducted three times.
- Please provide EDS maps of the wear tracks.
The study was complemented with SEM photos ‘of the wear track and EDS maps. The analysis was extended by description
Have you looked at the counter body worn surface? Have you observed material transfer? Please provide some images of the counter body.
Unfortunately, I do not have any images from the cross-section of the material. In the case of HDPE, a significant adhesive force and material transfer onto the counter-sample could be observed. However, for the remaining samples after the friction test, it was noticeable that the additives crumbled while the material transfer decreased, indicating a reduction in adhesive forces.
Comments on the Quality of English Language: Minor editing of English language required
The English language has been improved

Reviewer 4 Report
The research work “Influence of inorganic additives on the surface characteristics, hardness, friction 2 and wear behavior of polyethylene matrix composites”, is an interesting work it can be considered after a minor revision.
1. The abstract must have materials and combinations used followed by important results, that is missing.
2. Why do you choose titanium and boron nitride as fillers.
3. In the introduction you must mention the usage of inorganic fillers for different applications of PE polymer composites, see the following works related to that.
10.1002/pc.27320
4. The results needs to compared with latest works , that is missing.
5. Please only include the vital results in the conclusion part with possible future works
Author Response
Good Morning, I attach the answers and the manuscript.
a minor revision.
- The abstract must have materials and combinations used followed by important results, that is missing. The abstract looks good. Please include all significant numerical results.
The significant numerical results has been added
The aim of the research was to analyze the effect of inorganic additives on the tribological properties of the High-Density Polyethylene (HDPE) matrix composite surface. Titanium (Ti) and hexagonal boron nitride (hBN) were added in different mass fractions. The samples were produced by pressing a pre-prepared mixture of granules. The composite samples with the following mass fractions of additives were fabricated: 5% hBN, 10% hBN, 28% Ti-2% hBN, 23% Ti-7% hBN, and 20% Ti-10% hBN. An even distribution of individual additives' concentrations was confirmed. Observations of morphology, surface topography, hardness, and tribological measurements were conducted using reciprocating motion tests with the "pin on flat" and rotational tests with the "pin on disc" configuration. Subsequently, microscopic observations and measurements of the wear track profile were carried out. Additionally, geometry parameters of the contacting elastic body were calculated for various counter-samples. It was found that the Shore D hardness of samples containing Ti and hBN increases with the Ti content, while the coefficient of friction (COF) value decreases. The addition of hBN alone does not significantly affect the hardness, regardless of the ratio, while the COF increases with increasing hBN content. The COF value doubled with the addition of 10% hBN (COF = 0.22), whereas the addition of 90% Ti-10% hBN resulted in a decrease in the COF value to COF = 0.83. The highest hardness value was obtained for the sample containing 28% Ti-2% hBN (66.5), while the lowest was for the sample containing 10% hBN (63.2). The wear track analysis, including its height and width caused by deformation, was detected using a focal differentiation microscope and scanning electron microscopy. Additionally, EDS maps were generated to determine the wear characteristics of the composite.
- Why do you choose titanium and boron nitride as fillers.
The requiered nformation has been supplemented in 2.2. Materials.
The choice of titanium and boron nitride (hBN) as fillers for polyethylene matrix can be dictated by several factors. Both titanium and boron nitride have a beneficial impact on the hardness, strength, and tribological properties of the composite, including the sliding properties of hBN, which reduce friction and improve lubrication between surfaces. The addition of these fillers increases wear resistance, reduces friction and minimizes wear, as well as enhances high-temperature resistance. Furthermore, the chemical compatibility of titanium and boron nitride with polyethylene matrix facilitates their incorporation into the composite and maintains structural stability. Additionally, both titanium and boron nitride are non-toxic and can be safely used in contact with humans.
- In the introduction you must mention the usage of inorganic fillers for different applications of PE polymer composites, see the following works related to that.
The literature has been added.
[13].Wang, Q.; Wang, S.; Du, N.; Cheng, F.;Zhao, Q.; Li, X. Effects of HBN particle sizes on microstructure and perfor-mances of PEO coatings on 7075-T6 aluminum alloy. Zhongguo Youse Jinshu Xuebao/Chinese Journal of Nonferrous Metals. 2019, 29, 11, 2459 – 24701.
[14].Mahmoud, M., E.; Khalifa, M., A.; Youssef, M., R.; El-Sharkawy, R., M. Influence of MgO and ZnO as nano-additives on the mechanical, microstructural and thermal performance of high-density polyethylene. Journal of Applied Polymer Science. 2022, 139, 3115.
[15].Liyuan, Z.; Conglin, D.; Chengqing, Y.; Xiuqin, B.; Ye T. The role of graphene nanoplatelets in the friction reducing process of polymer. Polymer Composites. 2022, 43, 11, 8213 – 8227.
[16].Wu, Y.; Dong, C.; Yuan, C.; Bai, X.; Zhang, L.; Tian, Y. MWCNTs filled high-density polyethylene composites to im-prove tribological performance. Wear. 2021, 47718, 203776.
4. The results needs to compared with latest works , that is missing.
The studies found do not directly address the topic I am working on. I couldn't find any publications that apply hBN and Ti additives in an HDPE-based composite. Additionally, the use of such large grain sizes also complicates the task, as most researchers use additives with much smaller grain sizes.
- Please only include the vital results in the conclusion part with possible future works
The vital results were included.

Round 2
Reviewer 1 Report
The manuscript is well revised, and I agree to accept it.
Minor editing of English language required
Reviewer 2 Report
This manuscript has been carefully revised by considering the comments!
This review would like to recommend accepting this work!
Reviewer 3 Report
REVIEW
Review of the Manuscript MATERIALS-2458720-R1 entitled “Influence of inorganic additives on the surface characteristics, hardness, friction and wear behavior of polyethylene matrix composites” by Natalia Wierzbicka, RafaÅ‚ Talar, Karol Grochalski, Adam Piasecki, WiesÅ‚aw GraboÅ„, MiÅ‚osz WÄ™gorzewski and Adam Reiter to be considered for publication in the JOURNAL OF MATERIALS MDPI.
After revising the author's answers to the formulated questions, this reviewer suggests that the paper be accepted.